# Caffeine Consumption in Children: Innocuous or Deleterious? A Systematic Review

**DOI:** 10.3390/ijerph17072489

**Published:** 2020-04-05

**Authors:** Yeyetzi C. Torres-Ugalde, Angélica Romero-Palencia, Alma D. Román-Gutiérrez, Deyanira Ojeda-Ramírez, Rebeca M. E. Guzmán-Saldaña

**Affiliations:** 1Academic Area of Nutrition, Autonomous University of the State of Hidalgo, San Agustín Tlaxiaca 42086, Mexico; yeyetzi_torres@uaeh.edu.mx; 2Academic Area of Psychology, Autonomous University of the State of Hidalgo, San Agustín Tlaxiaca 42086, Mexico; 3Academic Area of Chemistry, Autonomous University of the State of Hidalgo, Mineral de la Reforma 42184, Mexico; 4Academic Area of Veterinary Medicine and Zootechnics, Autonomous University of the State of Hidalgo, Tulancingo de Bravo 43600, Mexico

**Keywords:** psychostimulant, caffeine, children, growth, development, health effects

## Abstract

Caffeine is the most consumed psychostimulant worldwide. Its use among children is controversial. Although it produces an increase in brain activity, it could hamper growth and development in young consumers. Therefore, the aim of this review was to recognize changes produced by caffeine in children under 12 years of age and to identify the relevant alterations and the conditions of their occurrence. A systematic review of the literature was carried out using PRISMA. Initially, 5468 articles were found from the EBSCO, ScienceDirect, PubMed, and Clarivate Analytics databases. In this review, were retained 24 published articles that met the inclusion criteria. The results obtained showed that caffeine consumption hampers children’s growth and development. In contrast, it supports the activation of the central nervous system and brain energy management.

## 1. Introduction

Caffeine belongs to the methylxanthines group and it is the most consumed psychostimulant (alkaloid) worldwide because it is present in coffee, which is the second most consumed liquid in the world, after water [1].

Coffee is the main source of caffeine in adults. This drink also contains phenolic acids, among which acylquinic acid and hydroxycinnamic acid stand out. These acids have been related to the decrease in the incidence of cardiovascular diseases, metabolic syndrome, and cancer [2]. Due to its potential pharmaceutical activity, coffee can be considered as a nutraceutical [3]. On the other hand, in studies conducted in adults, it has been observed that caffeine increases mental and physical performance [4].

Despite the beneficial effects of coffee in the adult population, its involvement in vulnerable groups as children remains unclear [5]. Half of the caffeine intake in children comes from cola drinks [6], and the energy drink consumption in this group has increased by 6% in the past decade [7,8,9].

Caffeine intake in children depends on the social context and culture [10]. For instance, supermarkets and restaurants, especially fast-food ones, are the places where caffeinated foods are the most easily found in Europe and the United States [11]. In Mexico, this intake also occurs in many socially significant scenarios, such as children’s parties, family gatherings, and school recess [12].

Caffeine consumption in the youth is a controversial topic. On one hand, the literature has shown evidence of alterations in children’s growth and development, such as iron absorption deficiencies and weight loss [13,14,15]. On the other hand, the physical, mental, attention-performance, and neuroprotective effects of caffeine have been demonstrated [13,15,16].

Seifert et al. [16] highlighted the cognitive and attention improvement effect in children when the alkaloid was supplied in small doses (under 2.5 mg/kg). Still, they concluded that caffeine intake in minors could lead to severe health consequences, especially when it was ingested together with other substances like taurine, as found in energy drinks.

Indeed, the physiological alterations produced by this substance depend on the amount ingested. Tieges et al. [17] and Ruxton et al. [18], described the amounts considered low (1 mg.kg.d), medium (3 mg.kg.d), and high (5 mg.kg.d) in children because these amounts differ in comparison with adults. The physiological effects of caffeine have been observed from moderate to high doses while no reports of organic changes were found at low doses.

Because the effects of caffeine are widely described in adults but not in children, a thorough review of the literature is necessary. Likewise, it is considered necessary to know if there are different effects produced depending on the dose and the source of intake.

For that reason, the aim of this systematic review of the literature was to investigate the main effects of caffeine in children. The benefits that could be obtained from this knowledge include providing accurate dietary guidance, considering caffeine as a risk factor to children’s health or a therapeutic agent in paedopsychiatric alterations, such as ADHD. By carrying out this systematic review, we aim to show the changes that caffeine produces in children’s metabolism, providing a physical, cognitive, and psychological panorama. This insight could favor the creation of strategies to limit or regulate the consumption of caffeine, depending on its negative or positive effects.

## 2. Materials and Methods 

This work was based on the PRISMA method (Figure 1) for systematic literature reviews, adapted from Urrútia and Bonfil [19]. The checklist developed by González et al. [20] was also followed.

### 2.1. Elegibility Criteria

For this review of the literature, the following concepts were proposed to obtain information:Caffeine is an alkaloid consumed in large amounts worldwide.It leads to physiological changes in consumers.These changes can be beneficial or detrimental to health.The changes produced are dose dependent.

Cross-sectional and longitudinal studies dealing with any aspect related to the physiological changes produced by caffeine were included. Only works conducted on humans and published in journals included in the Journal Citation Report (JCR) were considered.

### 2.2. Sources of Information

The search for information was conducted electronically in the digital repository of the Autonomous University of the State of Hidalgo, specifically in the digital collections.

Four search engines were found in this digital library: EBSCO, ScienceDirect, PubMed, and Clarivate Analytics. These search engines were selected due to their relevance since the publications in them possess a high impact factor. The search for information started on July 2nd, 2018 and concluded on December 20th in the same year.

### 2.3. Search for Information

The search terms, combination, and acronyms are described in Figure 2.

The lower limit of the search for information was year 1975 while year 2018 was the upper one. It was decided to start the search from articles published in 1975 since that was the year in which the first article on the effects of caffeine in children (with minimal brain dysfunction) was found. After a first look at the information, the search period was limited to 2008–2018, considering the articles published after 2008 were of higher relevance to this review.

A total number of 5,468 articles were found: 1530 in EBSCO; 1144 in PubMed; 1929 in ScienceDirect, and 865 in Clarivate Analytics, as shown in Figure 3.

Afterwards, experiments in animals, editorials, letters to editors, and brief communications were excluded, and 255 articles were retained. After a manual cross-check, duplicated reports were eliminated. Only articles published in scientific journals were included and 148 articles were obtained. From them, 93 articles were published from 2008 to date. Only 42 studies matched the inclusion criteria. They were subclassified according to the effect produced by caffeine as follows:Sixteen articles studied the effect of caffeine in children. Three articles were based on the concept of caffeine by consumers, two articles on the general activation state (arousal), two articles on physical activity, five articles on sleep cycle, and four articles on affective states.Seven articles were based on the secondary effects produced by caffeine in the treatment of respiratory disorders in newborns.Eighteen were review studies.

After a final selection, review articles were discarded and thus only the results of 24 articles were considered.

### 2.4. Study Selection

The inclusion criteria which led to the selection of the 24 papers from among the initially found articles were established according to the Participants, Interventions, Comparisons, Outcomes, and Studies (PICOS) design format adapted from Robinson et al. [21]:Participants: Children up to age 17 (upper limit) were considered. However, those studies with children under 12 years were preferred because there is evidence that sensitivity to caffeine is modified in adolescence and physiological, cognitive, and behavioral responses are different from those shown by children [22]. Both genders were considered, with or without pathologies, exhibiting a secondary physiological change after caffeine intake or administration at different doses.Interventions: Primary works on caffeine intake or administration and their effects on health were included. The type of survey used to determine the perception of caffeine, was also included.Comparisons and Outcomes: Studies using reliable evaluation instruments, measurement of effects of caffeine by image studies or physical tests sensitive to measurement of the alkaloid were included.Study design: All types of study design were included, with the exception of reviews and meta-analyses.

The exclusion criteria were:Studies not providing fundamental data on the effect of caffeine in children’s health.Studies in which caffeine was the result of a metabolic process and was not ingested or administered.Studies published before 2008.

### 2.5. Data Collection

An extraction database was designed considering the eligibility criteria of the articles. All the proposed methods in the research articles were analyzed and the database was completed with results, discussion, and conclusion analysis of each of the selected works. Once obtained, the information was synthesized, compared, and contrasted qualitatively between articles.

### 2.6. Result Synthesis

The experimental studies analyzed were catalogued in the topics on caffeine presented in Table 1. The years of publication are also shown.

## 3. Results

### 3.1. Caffeine Intake in Children

From the 23 studies meeting any of the inclusion criterion preestablished for analysis, those works explaining effects in children’s metabolism were selected (*n* = 20), whether positive (*n* = 11) or negative (*n* = 7). These were subdivided into categories, as shown in Table 2.

These categories are independently analyzed in the following paragraphs.

### 3.2. Concept of Caffeine Intake in the Population

The results of the perception that consumers have regarding caffeine can be seen in Table 3.

The experimental populations in the reviewed studies agreed in the unhealthy practice of caffeine intake. For example, Bucher and Siegrist [23] concluded that the analyzed population knew drinking caffeinated products could affect health. The participants classified energy drinks as the most harmful, followed by cola drinks. Wierzejska et al. [24] reported that 84.2% of the children in their study considered that kids should not consume energy drinks due to the potential health risks. Visram et al. [25] established that factors such as type of product, price, in-store location, and promotions were some of the main factors that lead to the consumption of these drinks.

### 3.3. Positive Effects

#### 3.3.1. Arousal

The articles related to the general activation state are shown in Table 4.

Barry et al. [26] expressed that there are some differences in the energy distribution patterns among children with respect to adults. For instance, they observed a global decrease of electroencephalogram (EEG) power in alpha and theta bands, reductions in power focal points of delta and beta bands, and an increase in focal points of alpha band in children’s EEGs, while no significant changes were observed in adults.

When the arousal system is affected, as in ADHD, caffeine proved to be useful to treat symptoms [27]. This study stated that the energy management alteration in this specific case could be the result of two specific circumstances: 1. a defect in the pathway to reach the central nervous system, and 2. an alteration in the arousal system. Both conditions improved when ADHD children consumed caffeine as compared to those who ingested a placebo.

#### 3.3.2. Effect of caffeine intake during physical activity

The analyzed studies on muscle effects in children are found in Table 5.

Turley et al. [28] observed that different doses of caffeine (defines as low = 1 mg/kg, moderate = 3 mg/kg, and high = 5 mg/kg) could lead to blood pressure and heart rate differences in each case. However, there is not enough support to assert these differences because the authors did not control the levels of caffeine consumption.

In the same line of investigation, Turley et al. [29] measured differences in physical endurance in relation with strength among children who consumed caffeine at different doses. Both parameters were higher at elevated doses of the alkaloid.

#### 3.3.3. Use of Caffeine in Respiratory Disorders among Newborns

The results of the works analyzed regarding the effect of caffeine in newborns are shown in Table 6.

Boia et al. [33] and Khurana et al. [35], indicated that caffeine is better than any other methylxanthine for the treatment of apnea of prematurity. According to the authors, this compound reduces the emergence of adverse cognitive, emotional, metabolic, and dietary effects, quite common under other treatments.

In addition, caffeine has been reported to be successful regarding the development of brain matter in premature infants weighing less than 1251 g at birth. Among this population, the treatment led to a significant change in white matter diffusion [31]. Similarly, the alkaloid improved the cognitive score in long-term follow-ups of one and two years [32]. Furthermore, motor improvement in children suffering consequences of apnea of prematurity who received treatment with this methylxanthine was demonstrated, compared to those children who did not receive it [34]. In addition, the children showed better scores in motor coordination, visuomotor integration, visual perception, and visuospatial organization [36]. Finally, the administration of caffeine benzoate caused fewer adverse post-intubation effects (laryngospasm, upper airway obstruction, and oxygen desaturation) in children after an adenotonsillectomy by obstructive sleep apnea [30].

### 3.4. Negative Effects

#### 3.4.1. Alterations in Sleep Cycle

The results of the analyzed studies are presented in Table 7.

The results of the studies in this section show that caffeine could lead to alterations in the children’s sleep cycle. Nevertheless, the intake of the alkaloid is related to other kinds of alterations, such as asthma, gastritis, allergies, sleep disorders, and enuresis [37,38].

Electronic devices are also detrimental to the sleep duration. Calamaro et al. [39] reported that the relationship between caffeine and these gadgets was negative since the latter decreased total sleeping time by 15 minutes. In this study, the sleep hygiene of children who consumed caffeine (1–5 cups / cans of caffeinated beverages) was compared against that of children who did not consume it. The body mass was not taken into account.

Similar results were found in a study by Katz et al. [40] where the intake of substances as caffeine to minimize the effects of daytime drowsiness made children spend an extra 4 minutes trying to fall asleep when compared with those that did not consume any substances. In this case, the caffeine indicator was its presence in urine samples analyzed through a qualitative gas chromatography/mass spectroscopy (GC/MS) screen. An analysis of the level of consumption was not performed, and caffeine samples were only compared with those that did not contain caffeine.

Finally, Watson et al. [41] observed that chronic changes in the sleeping pattern as a consequence of the alkaloid could predispose consumers to some psychological alterations like depression, anxiety, and psychosomatization. To assess the level of caffeine consumption, the authors used a self-report caffeine food frequency questionnaire, which calculated caffeine values in mg/day.

#### 3.4.2. Relationship with Affective States

The results of this section are presented in Table 8.

Luebbe and Bell [43] observed that children who frequently (three or more times per week) consumed caffeine exhibited greater emotional lability and depression compared to adolescents with a higher body mass. A publication on the connection between caffeine consumption and child depression [44] clarified that a high intake of cola and energy drinks in children can be directly related to depression among this population. The authors reported that children consumed on average 15.24 mg/kg of caffeine per week and teenagers consumed 13.82 mg/kg per week, but the number of days of intake were different in children (3.69 days) and adolescents (4.89 days). This variation could be attributed to the difference in body mass. On the other hand, the observed psychological effects were present with a mean of 11.36 in children against 8.6 in adolescents.

Furthermore, Whalen et al. [42] observed that children with depression consume higher amounts of the alkaloid since it reduces the negative symptoms related to the disorder. The caffeine ingested was measured considering the number of caffeinated beverages per day but not the dose.

Finally, Temple et al. [22] reported that children who did not consume caffeine several times per week (more than three) and did only for the study purposes (1 and 2 mg/kg) showed greater risk behaviors and impulsiveness, and they also sought pleasant sensations.

## 4. Discussion

In the literature reviewed for this study, some methodological problems were evidenced:The position (if there were positive or negative effects) of the studies on caffeine consumption in minors was unclear.The analyzed studies do not compare positive and negative effects of caffeine.The studies do not consider the different metabolic processes in children and adults.There were few longitudinal studies, so the long-term effect of caffeine could not be known.The age limits between the studies were poorly specified and most cases did not differentiate children from adolescents.Few studies analyzed considered the different doses of caffeine, so the comparison of effects according to dose could not be performed.

The present review tried to avoid the previous misunderstandings and methodological complications. The separation of references in two groups allowed for a better analysis of the information. This review emphasizes that the alkaloid is metabolized differently in children and adults and some physiological processes considered beneficial to adults may not be desirable in children. Then, a more accurate orientation towards the effects of caffeine in children’s metabolism is possible.

Furthermore, the results of this research showed that consumers possess little to no knowledge on the effects produced by caffeine in the organism, although they assume there is a negative effect on health. In all cases, the perception of caffeine was clearly negative and independent of the intake recommendations. Evidently, the population knows the possible health risks these products possess, although this perception does not agree with current advertisements.

If the classification proposed by Tieges et al. [17] and Ruxton et al. [18] is exclusively taken into account, children with a caffeine intake of 1 mg.kg.day would have a low intake, 3 mg.kg.day a medium intake and 5 mg.kg.day a high intake. Although no correlation was found between doses and effects in the literature analyzed, evidence found in this research shows that the deleterious effects of caffeine may occur, above all, at moderate and high doses. However, the authors of this review consider that the effects on the sleep cycle that low doses might cause should not be underestimated.

Understanding that caffeine in moderate and high doses is deleterious to healthy children can be useful to provide a real nutritional guidance to the youth. Additionally, it could help in the creation of advertising strategies in order to not compromise children’s health. For example, in nutrition guidelines it could be proposed that caffeine is “no consumption” or “moderate consumption” for children.

From the systematic review of the current literature on the physiological changes produced by caffeine in children, the evidence indicates that caffeine could be recommended for the youth with especial conditions (ADHD and apnea of prematurity) but not suggested for healthy children, especially in moderate and high doses that reportedly cause physiological alterations.

Considering the published studies, two main groups were established according to the effects (positive or negative) of caffeine consumption in children.

The positive effects of caffeine found in children’s metabolism were: Improvement of energy distribution in the central nervous system. Regarding this factor, the studies analyzed [26,27], did not take into account the weight of the children, so they did not consider the consumption level, whether high, medium or low. Besides, the studies did not include other populations or physiological conditions; even the symptoms that were reduced as a consequence of a better energy management in the brain were not specified. That is the reason why further investigation considering the consumption level is considered necessary.Improvement of physical performance. Caffeine could improve the physical performance of children in a dose-dependent manner. Nevertheless, different doses are difficult to control, and it must be considered that children are more sensitive to the alkaloid, although the cause is unknown.Improvement of the respiratory function. These effects were especially positive when there is an alteration in the brain or respiratory dynamic, as ADHD or apnea of prematurity. In those cases, caffeine could be a first-line therapeutic agent or an adjuvant of the medical treatment. Apnea of prematurity is a medical condition frequently present in newborns under 32 weeks of gestation. The treatment of respiratory disorders using caffeine has been reported to be effective for the cognitive performance in the short and long term.

Furthermore, the negative effects of the alkaloid in moderate and high doses were (1) alterations in the sleep cycle and (2) the affective states. In the first case, children’s metabolism is increased at night. Their sleep cycle demands stability and consistency as well as more hours in bed than adults [39]. As observed, any alteration in the sleep cycle indirectly compromises the adequate mental and physical development of children. The ingestion of caffeine is one of the leading factors interfering with these processes. It would be important to know if there are variations depending on different doses. 

Alterations of affective states as anxiety and depression produced by caffeine are directly proportional to the amount ingested. Since children’s body mass is lower than that of adults, they consume more caffeine but in fewer products [43]. This effect seems to be more harmful in the case of affective states. Children’s emotional health seems to be one of the aspects affected by caffeine intake and its dosage influence. Therefore, this information may suggest that children are more sensitive to the effects of the alkaloid, although the cause remains unclear.

Finally, caffeine affects the way in which energy is distributed in the central nervous system. Additionally, metabolic effects, such as changes in heart rate, blood pressure, and the ventilatory function should also be taken into account. These effects occur mostly among young people, probably as a result of the dosage problem that has been discussed throughout this manuscript. Therefore, this factor should be included in a paedopsychiatric and medical evaluation.

However, in special cases as those of children who suffer pathologies that compromise the brain dynamic and respiratory function, caffeine could be considered as an adjuvant of the treatment. Still, the treatment with caffeine could compromise the children’s growth and development if the symptoms persist. Therefore, it can be said that caffeine is innocuous only under special conditions

## 5. Conclusions

The effects of caffeine in children can be divided into positive and negative. The first ones are found at cognitive level (elevated short-term arousal and increased motor activity, perception, and intelligence in the long term) and improved physical performance both in aerobic and anaerobic conditions. On the other hand, the negative effects include changes in the sleep cycle, which could indirectly alter the weight and growth of children, and greater sensitivity to the alkaloid at an emotional level (anxiety and depression).

### Limits, Advantages and Potential Applications

Due to this systematic review provides accessible evidence about the effects that caffeine produces in the children’s organism, it could cater knowledge that could guide consumption behavior in this particular population.

Likewise, in this review it was evidenced that the general population considered that the intake of this alkaloid could produce negative effects on the health of school children. Therefore, this material could be useful to provide scientific evidence on the effects it produces and reinforce the general knowledge that the population has about caffeine.

The investigations analyzed in this systematic review of the literature did not take into account the different consumption doses of their study population. Thus, it was difficult to establish a correlation between the consumption doses and the observed effects. Future research is considered necessary to correlate doses with effects.

Despite the fact that this review emphasizes that caffeine is metabolized differently in children and adults, more research is considered necessary on the metabolic effects that occur in children compared to adults, for example in the change of enzyme markers.

In the same order of ideas, further investigation of the effects of caffeine in the field of paidopsychiatry is considered necessary, since in this review, a lack of consensus was observed on whether the effects it produces in these pathologies could be considered positive or negative.

## Figures and Tables

**Figure 1 ijerph-17-02489-f001:**
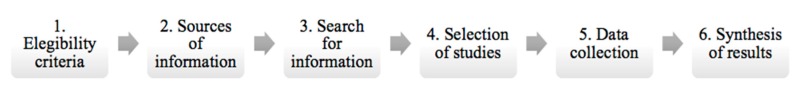
Steps for a systematic review of literature.

**Figure 2 ijerph-17-02489-f002:**
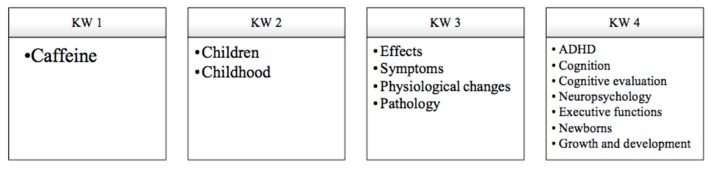
Search criteria for information and keywords (KW).

**Figure 3 ijerph-17-02489-f003:**
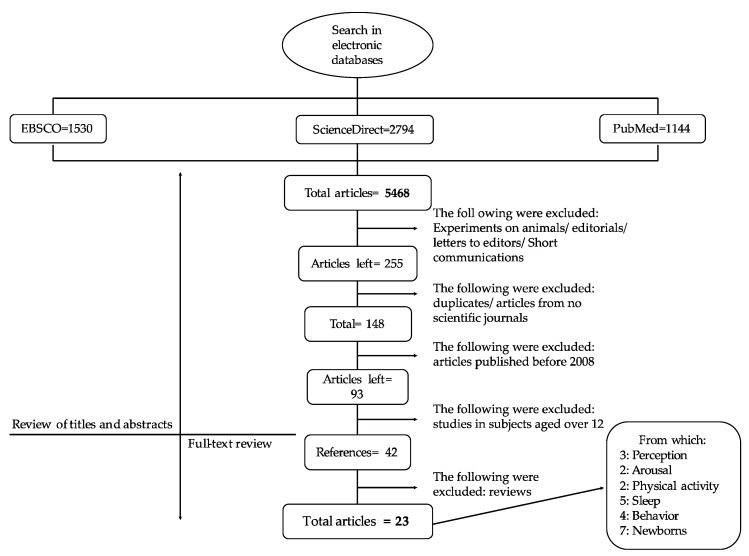
Process of systematic review of the literature.

**Table 1 ijerph-17-02489-t001:** General summary of the characteristics of the experimental studies analyzed.

Characteristic	Number of Studies (*n* = 23)
**Publication Year:**	**23:**
2008–2009	5
2010–2011	5
2012–2013	2
2014–2016	5
2016–2017	5
2018	1
**Topics Analyzed:**	**23:**
Concept	3
Arousal	2
Exercise	2
Sleep	5
Behavior	4
Newborns	7

**Table 2 ijerph-17-02489-t002:** Study categories of effects produced by caffeine in children.

Positive Effects (*n* = 11)	Negative Effects (*n* = 9)
Arousal (*n* = 2)	Sleep disorders (*n* = 5)
Physical activation (*n* = 2)	Affective disorders (*n* = 4)
Treatment of respiratory disorders in newborns (*n* = 7)	

**Table 3 ijerph-17-02489-t003:** Analysis of studies on caffeine intake perception, according to the PICOS ^1^ method.

Study	Participants	Interventions	Comparisons	Outcomes	S. Design
Bucher and Siegrist (2015) [23]	100 children with one parent (*n* = 100 couples); ages 7–10	Beverage-sorting task	Product classification in healthy and unhealthy	Negative perception of caffeine	Cross- sectional
Wierzejska et al. (2016) [24]	329 children: ages 11–13	Food frequency questionnaire	Personal interviews	Negative perception of caffeine	Face-to-face interview
Visram et al. (2017) [25]	37 children, 2 groups: ages 10–11 (*n* = 20) and ages 13–14 (*n* = 17)	Semi-structured focus groups	Personal opinions	Negative perception of caffeine	Qualitative, based on data

^1^ PICOS = P: participants; I: interventions; C: comparisons; O: outcomes; S: study design.

**Table 4 ijerph-17-02489-t004:** Description of analyzed studies on arousal, according to the PICOS ^1^ method.

Study	Participants	Interventions	Comparisons	Outcomes	S. Design
Barry et al. (2009) [26]	30 children; ages 8–13	Caffeine administration: 80 mg (1.3–3.3 mg/kg/day)	Caffeine vs. placebo: electrodermal activity and EEG ^2^ changes	Caffeine increases arousal	Randomized double blind placebo control cross-over trial
Barry et al. (2012) [27]	18 ADHD children; ages 8–13.	Caffeine administration: 80 mg caffeine	Caffeine vs. placebo: electrodermal activity	Caffeine increases arousal	Randomized double blind placebo control cross-over trial

^1^ PICOS = P: participants; I: interventions; C: comparisons; O: outcomes; S: study design. ^2^
*EEG = Electroencephalogram*.

**Table 5 ijerph-17-02489-t005:** Description of analyzed studies on physical activation, according to the PICOS ^1^ method.

Study	Participants	Interventions	Comparisons	Outcomes	S. Design
Turley, et al. (2008) [28]	40 children; ages 7–9	Caffeine administration: 1, 3, and 5 mg/kg/day	Caffeine vs. placebo: ergometry, blood pressure, and cardiac frequency 60 minutes before exercise	Caffeine increases BP ^2^ and reduces HR ^3^	Randomized double blind placebo control counter-balanced
Turley et al. (2014) [29]	26 children (male); ages 8–10	Caffeine administration: 1, 3, and 5 mg/kg/day	Caffeine vs. placebo: static handgrip and ergometry	Caffeine provides greater strength and physical performance	Randomized double blind placebo control counter-balanced

^1^ PICOS = P: participants; I: interventions; C: comparisons; O: outcomes; S: study design. ^2^ BP = Blood pressure. ^3^ HR = Heart rate.

**Table 6 ijerph-17-02489-t006:** Description of analyzed studies on respiratory disorders, according to the PICOS ^1^ method.

Study	Participants	Interventions	Comparisons	Outcomes	S. Design
Khalil et al. (2008) [30]	72 children in 2 groups: experimental *n* = 36 and control; ages 2.5–12	Caffeine benzoate administration: 20 mg/kg IV	Caffeine vs. placebo: presence of adverse respiratory events post-extubation	Caffeine improves respiratory function	Randomized double blind placebo control
Doyle et al. (2010) [31]	70 preterm infants aged 10 days	Caffeine citrate administration: 20 mg/kg	Caffeine vs. placebo: IRM differences	Caffeine improves development of white matter	Randomized placebo control
Gray et al. (2011) [32]	287 infants under 30 WOG^2^ suffering apnea of prematurity	Caffeine citrate administration: group 1 = 5 mg/kg; group 2 = 20 mg/kg	Low dose vs. high dose: development and cognition study 1 year after; temper study 2 years after.	Caffeine in apnea of prematurity does not affect cognition nor behavior	Multicenter randomized controlled trial
Boia et al. (2014) [33]	84 pre-term infants < 32 WOG ^2^ and < 1500 g suffering apnea of prematurity	Caffeine citrate administration: 5 mg/kg/day. Theophylline administration: 3 mg/kg/day	Caffeine vs. theophylline: adverse effects	Caffeine causes fewer adverse effects tan theophylline in apnea of prematurity	Retrospective analysis
Doyle et al. (2014) [34]	1433 infants suffering apnea of prematurity and who developed cerebral palsy	Caffeine citrate administration at therapeutics doses; test battery for movement evaluation; Wechsler for preschoolers; primary scale of intelligence III	Caffeine vs. placebo	Caffeine leads to fewer coordination disorders	Randomized and retrospective controlled trial
Khurana et al. (2017) [35]	240 infants aged 18–24 months and corrected apnea of prematurity	Caffeine citrate administration: 20 mg/kg/day. Theophylline administration: 5 mg/kg/day	Caffeine vs. theophylline: cognitive performance	Caffeine in apnea of prematurity improves cognitive performance and reduces motor deficiencies	Randomized and retrospective controlled trial
Mürner-Lavanchy et al. (2018) [36]	870 children; age 11.	Caffeine citrate administration 20 mg/kg	Caffeine vs. no caffeine: intelligence,	Caffeine in apnea of prematurity	Randomized double blind

^1^ PICOS = P: participants; I: interventions; C: comparisons; O: outcomes; S: study design. ^2^ WOG = Weeks of gestation.

**Table 7 ijerph-17-02489-t007:** Description of analyzed studies on sleep disorders, according to the PICOS ^1^ method.

Study	Participants	Interventions	Comparisons	Outcomes	S. Design
Warzak et al. (2011) [37]	201 children; ages 5–12	Caffeine intake: 52–109 mg/day	Evaluation of enuresis and sleep history	Caffeine reduces total sleeping time	Cross-sectional
Calhoun et al. (2011) [38]	77 children with excessive day drowsiness; ages 5–12	Caffeine intake questionnaire	Caffeine vs. no caffeine ^2^: polysomnography	Caffeine is not associated to excessive day drowsiness	Cross-sectional
Calamaro et al. (2012) [39]	625 children; ages 6–10	Caffeine intake: 1–5 cans of soda or cups of coffee per day	Low caffeine intake vs. high caffeine intake by frequency questionnaires	Caffeine produces 15 minutes less of sleep per night	Cross-sectional
Katz, et al. (2014) [40]	210 children; ages 9–15	Drug testing	Caffeine detected or undetected by drug testing	Caffeine increases start of sleep by 4 minutes	Retrospective analysis
Watson et al. (2017) [41]	309 children; ages 8–12	Caffeine intake: 0–151 mg/day	Caffeine versus no caffeine	Caffeine is related to sleep disorders	Cross-sectional

^1^ PICOS = P: participants; I: interventions; C: comparisons; O: outcomes; S: study design. ^2^ Other variables besides caffeine were considered in this study.

**Table 8 ijerph-17-02489-t008:** Description of analyzed studies on affective states, according to the PICOS ^1^ method.

Study	Participants	Interventions	Comparisons	Outcomes	S. Design
Whalen et al. (2008) [42]	53 children and adolescents (30 with major depressive disorder); ages 7–17	Caffeine intake questionnaires; structured diagnosis interviews	Caffeine vs. no caffeine: structured diagnostic interviews differences	Higher caffeine intake in major depressive disorder	Clinical evaluation longitudinal study
Luebbe and Bell (2009) [43]	135 children aged 10–12; 79 adolescents aged 15–17	Frequency of consumption questionnaire of caffeine; depression inventory for youth; treatment of anxiety inventory for children	Low and high caffeine consumption: mood differences	Young children who consume higher doses proportionally to weight are more sensitive to caffeine	Cohorts
Benko et al. (2011) [44]	51 children; aged 9–12	Nutrition-behavior inventory; depression inventory for youth; child behavior checklist	Low vs. high caffeine intake: mood differences	Caffeine increases depressive symptoms	Longitudinal

^1^ PICOS = P: participants; I: interventions; C: comparisons; O: outcomes; S: study design.

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
