# Peer review of "Caffeine Consumption in Children: Innocuous or Deleterious? A Systematic Review"

_ijerph, 2020, doi:10.3390/ijerph17072489_

Round 1
Reviewer 1 Report
An added fragment about the toxic dose of caffeine for children (line 52-56) is not satisfactory. Apart from opinions of other experts authors of this article should give their point of view in this issue. For me dose 3 mg/kg/day as a medium intake, and 2.5 mg/kg/day as a toxic intake is not understandable
Author Response
Response to Reviewer 1 Comments
Point 1: An added fragment about the toxic dose of caffeine for children (line 52-56) is not satisfactory. Apart from opinions of other experts authors of this article should give their point of view in this issue. For me dose 3 mg/kg/day as a medium intake, and 2.5 mg/kg/day as a toxic intake is not understandable.
Response 1: The sentence (lines 58-60) “In the same sense, Sifert et al. [19], indicated that the toxicity threshold is 2.5 mg.kg.d in children under 12 years”, was removed because it produced confusion, limited the fluidity of the text and its exclusion does not contradict the meaning of the investigation.
Reviewer 2 Report
The authors have improved the paper. Only one suggestion: The authors should highlight in the Conclusion limits, advantages and potential applications of this research
Author Response
Response to Reviewer 2 Comments
Point 1: The authors have improved the paper. Only one suggestion: The authors should highlight in the Conclusion limits, advantages and potential applications of this research.
Response 1: The section Limits, advantages and potential applications, was added (lines 386-403):..“Due to this systematic review provides accessible evidence about the effects that caffeine produces in the children's organism, it could cater knowledge that could guide consumption behavior in this particular population. Likewise, in this review it was evidenced that the general population considered that the intake of this alkaloid could produce negative effects on the health of school-children. Therefore, this material could be useful to provide scientific evidence on the effects it produces and reinforce the general knowledge that the population has about caffeine. The investigations analyzed in this systematic review of the literature did not take into account the different consumption doses of their study population. Thus, it was difficult to establish a correlation between the consumption doses and the observed effects. Future research is considered necessary to correlate doses with effects. Despite the fact that this review emphasizes that caffeine is metabolized differently in children and adults, more research is considered necessary on the metabolic effects that occur in children compared to adults, for example in the change of enzyme markers. In the same order of ideas, further investigation of the effects of caffeine in the field of paidopsychiatry is considered necessary, since in this review, a lack of consensus was observed on whether the effects it produces in these pathologies could be considered positive or negative…”
Reviewer 3 Report
General comment: The review article entitled “Caffeine consumption in children: innocuous or 2 deleterious? A systematic review.” summarize the possible effect of caffeine in children. This is an interesting study. Some minor corrections are required for the improvement of the manuscript.
Abstract: The Abstract is well written presents the background and the aim of the study.
-Authors could further describe the methods used (e.g. how many studies initially used, databases used) by adding 1-2 lines.
Introduction: The introduction section is well-written and adequately presents the possible positive or negative effects of caffeine in children and the need for further investigation.
Materials and Methods: The methods used fully described. The methodology session is analytical and adequately presented.
Results and Discussion: Authors presents adequately the results of the selected studies, using useful different Tables in each situation.
Authors fully discuss the results of the study and divide the possible positive or negative effects.
- Could authors define it is proposed to be caffeine consumption in nutrition guidelines for “no consumption” or moderate consumption for children?
-Could authors further discuss the caffeine dose in which no negative effects can be presented in children?
Conclusion: The conclusion is adequate and summarizes the main text.
Bibliography/References: The references used by the authors cover adequately the relative scientific field and the aims of the study.
Author Response
Response to Reviewer 3 Comments
Point 1: Abstract: Authors could further describe the methods used (e.g. how many studies initially used, databases used) by adding 1-2 lines.
Response 1: The following data was included (lines 19-22):..”A systematic review of the literature was carried out using PRISMA. Initially, 5,468 articles were found from the EBSCO, ScienceDirect, PubMed, and Clarivate Analytics databases. In this review, were retained 24 published articles that met the inclusion criteria”...
Point 2: Results and discussion: Could authors define it is proposed to be caffeine consumption in nutrition guidelines for “no consumption” or moderate consumption for children?
Response 2: The next paragraph (lines 332-334), was adecuated for the reviewer suggestion:..”Understanding that caffeine in moderate and high doses is deleterious to healthy children can be useful to provide a real nutritional guidance to the youth. Additionally, it could help in the creation of advertising strategies in order to not compromise children’s health. For example, in nutrition guidelines it could be proposed that caffeine is “no consumption” or “moderate consumption” for children”…
Point 3: Results ans discussion: Could authors further discuss the caffeine dose in which no negative effects can be presented in children?
Response 3: The next paragraph about doses, was added to the discussion (lines 323-329):..”If the classification proposed by Tieges et al. [17] and Ruxton et al. [18] is exclusively taken into account, children with a caffeine intake of 1 mg.kg.day would have a low intake, 3 mg.kg.day a medium intake and 5 mg.kg.day a high intake. Although no correlation was found between doses and effects in the literature analyzed, evidence found in this research, shows that the deleterious effects of caffeine may occur, above all, at moderate and high doses. However, the author of this review consider that the effects on the sleep cycle that low doses might cause should not be underestimated”...
This manuscript is a resubmission of an earlier submission. The following is a list of the peer review reports and author responses from that submission.
Round 1
Reviewer 1 Report
The reviewer thanks the author's for their work on the manuscript. The topic of harms and benefits of caffeine consumption in children is an important topic. To enhance your manuscript, I recommend the following changes be made:
Introduction:
Line 43: Author's state that caffeine consumption leads to serotonin syndrome however this is in combination with antidepressants or other serotonergic agents. Caffeine actually decreases serotonin with chronic consumption. Also, serotonin syndrome is not particularly relevant to growth and development.
Line 47: Replace "autonomous" with "autonomic"
Line 47: This is a review article and the author did not "find". This needs to be re-worded. This statement is also a misrepresentation of what is described in the article. It is stating caffeine in combination with other serotonergic agents increase the risk of serotonin syndrome. This must be addressed.
Paragraph 5 (intro) - Needs to be looked at completely. It currently does not make sense.
The entire introduction lacks flow and needs to be re-written appropriately.
The aims of the study are not clear that you are performing a systematic review. This needs to be clarified.
Line 70: Spelling error.
Line 96: Punctuation
Line 110: Spelling ("disfunction" should be "dysfunction"
Results/methods
Authors state that you excluded studies with participants greater than 12 years of age however a number of studies have been included with participants in this age group. Please clarify.
Lines 213-215: Reference for this fact? There appears to be confusion here as well with it suggested that caffeine is an agonist at adenosine receptors however it is an antagonist. Please clarify.
Author's have included in utero exposure in their results however this does not form part of the inclusion criteria.
Discussion
Only limited discussion is provided and is mixed in with the results section. This should be in either one or the other section.
Results and discussion list improved cognitive performance as a negative whereas this is actually a positive? Please clarify.
The discussion really lacks depth when discussing reasons why children may be more susceptible to the effects of caffeine. More depth needed e.g. differences in metabolising enzymes?
Overall
English grammar and spelling check is needed throughout the manuscript
Overall the aims and concepts in the manuscript feel disjointed and a lot of casual language is used. This needs to be fixed.
Reviewer 2 Report
The article is an interesting and comprehensive summary of current knowledge on the influence of caffeine on children's body.
I have no significant objections, but some sections need to be corrected.
- In the Introduction (line 27), the sentence "Caffeine is the most consumed psychostimulant due to its simple preparation"???? is incomprehensible. After all, consumers do not use pure caffeine - as such - to prepare beverages !
- In the Introduction (lines 30-32), the sentence "Caffeine consumption has been related to beneficial effects in health among the adult population. For instance, it reduces the risk of type 2 diabetes, cardiovascular disease …" is not true. It's not the effect of caffeine, but coffee !
The beneficial effect of coffee on health, currently found in the literature, is most probably due to the presence of polyphenols, not caffeine. Studies show that decaffeinated coffee also has a beneficial effect. The publication quoted here also concerns coffee.
- In the Introduction (line 61) - should it not be "in children under the age of 12" instead of "in infants under the age of 12” - because infants are children in the first year of life
What is more, how can authors explain the fact that for children, a caffeine dose of 3 mg/kg/day means a mild intake, and a high intake is 5 mg/kg/day, since the threshold for caffeine toxicity in children is 2.5 mg/kg/day? Such a comment of authors must be added
- In Materials and Methods (line 95) the SCOPUS database is given, while in line 114 the ScienceDirect ?
- In Study selection (line 151, 157) The exclusion criteria were… information "studies including participants above 12” needn’t have been provided since in Inclusion criteria the following has been indicated: "children up to age 12 (upper limit)".
- in tables No. 5 and No. 6, abbreviations not previously explained (BP, CF, WOG) are used
- In Discussion (negative effects - line 379) it should be "reduced cognitive performance "instead of "improved cognitive performance" - which is consistent with the table No. 9 ???
Reviewer 3 Report
The subject is very interesting and answer to a question in line with the actual question of scientific research. This should marked in the Conclusion. Moreover the paper is well written and organized.
In Introduction the authors should add some introductive lines on bioactive compounds and nutraceuticals and mention related references such as:
Durazzo, A.; Lucarini, M.; Souto, E.B., Cicala, C.; Caiazzo, E.; Izzo, A.A., Novellino, E.; Santini, A. Polyphenols: a concise overview on the chemistry, occurrence and human health. Phyt. Res. 2019, 33, 2221-2243.
Santini, A.; Novellino, E. Nutraceuticals-shedding light on the grey area between pharmaceuticals and food. Expert. Rev. Clin. Pharmacol. 2018, 11, 545–547.